# Three-dimensional localization of nanoscale battery reactions using soft X-ray tomography

Young-Sang Yu[1,2], Maryam Farmand[1], Chunjoong Kim[2,3], Yijin Liu [4], Clare P. Grey[5,6], Fiona C. Strobridge[5], Tolek Tyliszczak[1], Rich Celestre [1], Peter Denes[1], John Joseph[7], Harinarayan Krishnan [8], Filipe R.N.C. Maia[9], A.L.David Kilcoyne [1], Stefano Marchesini[1], Talita Perciano Costa Leite [8], Tony Warwick[1], Howard Padmore[1], Jordi Cabana [2] & David A. Shapiro [1]

Battery function is determined by the efficiency and reversibility of the electrochemical phase transformations at solid electrodes. The microscopic tools available to study the chemical states of matter with the required spatial resolution and chemical specificity are intrinsically limited when studying complex architectures by their reliance on two-dimensional projections of thick material. Here, we report the development of soft X-ray ptychographic tomography, which resolves chemical states in three dimensions at 11 nm spatial resolution. We study an ensemble of nano-plates of lithium iron phosphate extracted from a battery electrode at 50% state of charge. Using a set of nanoscale tomograms, we quantify the electrochemical state and resolve phase boundaries throughout the volume of individual nanoparticles. These observations reveal multiple reaction points, intra-particle heterogeneity, and size effects that highlight the importance of multi-dimensional analytical tools in providing novel insight to the design of the next generation of high-performance devices.

[1] Advanced Light Source, Lawrence Berkeley National Laboratory, Berkeley, CA 94720, USA. [2] Department of Chemistry, University of Illinois at Chicago, Chicago, IL 60607, USA. [3] Department of Materials Science and Engineering, Chungnam National University, Daejeon, Chungnam 305-764, South Korea. [4] Stanford Synchrotron Radiation Lightsource, SLAC National Accelerator Laboratory, Menlo Park, CA 94025, USA. [5] Department of Chemistry, University of Cambridge, Lensfield Road, Cambridge CB2 1EW, UK. [6] Department of Chemistry, Stony Brook University, Stony Brook, NY 11794, USA. [7] Engineering Division, Lawrence Berkeley National Laboratory, Berkeley, CA 94720, USA. [8] Computational Research Division, Lawrence Berkeley National Laboratory, Berkeley, CA 94720, USA. [9] Department of Cell and Molecular Biology, Uppsala University, Husargatan 3, 75124 Uppsala, Sweden. Correspondence and requests for materials should be addressed to J.C. (email: jcabana@uic.edu) or to D.A.S. (email: dashapiro@lbl.gov)

Techniques capable of analyzing chemical states at high spatial resolution are essential for elucidating the complex phenomena at the nanoscale that underpin materials' properties. For example, battery function is determined by the efficiency and reversibility of the electrochemical phase transformations at solid electrodes, creating the need to accurately define relationships between chemistry, mechanics, and morphology[1, 2]. Conventional X-ray imaging methods are well suited to probe chemical states in bulk matter, but they are also limited in spatial resolution to a few tens of nanometers by the X-ray optics[3–5]. Furthermore, bulk X-ray diffraction can unambiguously differentiate between two-phase and metastable single-phase delithiation pathways, but it cannot map heterogeneities in the spatial distribution of such states[6]. In turn, electron-based techniques achieve very-high spatial resolution[7–9] and can provide three-dimensional (3D) quantification of the chemical state[10], but they also suffer from diffraction contrast effects and non-linearities for material thicknesses greater than the mean-free-path of inelastic scattering. Soft X-ray ptychography has recently narrowed the gap in spatial resolution while retaining high sensitivity to chemical states and penetration through functional volumes of matter[11, 12]. If data are only collected along one two-dimensional (2D) projection, the analysis of complex systems becomes problematic because of the likelihood of overlapping material with differing chemical components[3, 4]. This problem is readily solved by the use of X-ray based computed tomography, but the quantification of chemical states in three dimensions by conventional methods comes with limited spatial resolution, which is currently, at best, 30 nm[3, 13–15].

Here, we have combined soft X-ray ptychographic imaging and computed spectro-tomography to determine the 3D morphology and oxidation states of transition-metal cations in agglomerated cathode nanoparticles of lithium iron phosphate (LiFePO$_4$) at 11 nm 3D spatial resolution. The measured absorption at each voxel and X-ray photon energy is converted to optical density (OD) and used for computing quantitative 3D chemical composition. We investigate the complex correlation between chemical phase distribution and morphology in single nano-plates of LiFePO$_4$, a material that epitomizes the fundamental nature of intercalation chemistry that enables electrodes for high energy density Li-ion batteries[16, 17]. The mechanism of transformation of LiFePO$_4$ is one of the most intensely studied reactions in battery chemistry. While the reaction proceeds through a first-order transition in equilibrium[16, 18–20], under certain kinetic conditions, metastable pathways based on solid solutions have been observed[21–24]. These pathways bypass penalties in coherency strain due to the co-existence of phases in one particle, both enabling completion of the reaction and faster kinetics. The exact conditions that determine these pathways and, more generally, how electrochemical transformations can occur within single particles of battery electrodes are still widely debated topics. Our approach enabled both direct observation of the static internal chemical structure within crystals as small as 20 nm in their smallest dimension and the evaluation of correlations of the state of charge with particle size among a statistically significant number of particles.

## Results

**Sample synthesis and 3D chemical mapping.** LiFePO$_4$ nano-plates ($100 \times 80 \times 20$ nm$^3$) were electrochemically delithiated in a Li metal half-cell until 50% of the total amount of lithium was extracted, based on coulometric analysis of the cell response (see Methods and Supplementary Figs. 1–2). The delithiation was conducted at a slow rate to maximize reaction homogeneity[25] across the electrode and minimize the formation of metastable states that could relax during the harvesting of the particles[22].

Coulometry is an adequate method to control the average composition of the electrode due to the absence of side reactions at the potentials of operation[26]. Indeed, bulk X-ray diffraction of the partly delithiated sample confirmed the co-existence of two phases (Supplementary Fig. 3). The position of the diffraction peaks of one phase were in agreement with Li$_\alpha$FePO$_4$, where $\alpha$ was slightly smaller than 1, consistent with small domains of Li solubility observed in previous studies[16, 27, 28]. In contrast, the second phase showed peak positions comparable to FePO$_4$, consistent with reports that the solubility of Li on the Li-poor end of the phase diagram, $\beta$ in Li$_\beta$FePO$_4$, is very small[27]. Analysis of the relative intensities using methodologies of analysis in the literature[29] confirmed the presence of these two phases at ~50% ratio. The large facets of the plates correspond to the $ac$ crystallographic plane with the long axis parallel to $c$[12, 30].

Tomographic data from over 100 harvested particles were collected near the Fe $L_3$ edge at 708.2 and 710.2 eV (Fig. 1a and Supplementary Movie 1), which correspond to the maxima of the absorption resonances for LiFePO$_4$ (Fe$^{2+}$) and FePO$_4$ (Fe$^{3+}$), respectively, as shown in Supplementary Fig. 5 and in the literature[31–33]. The 3D resolution of 11 nm was confirmed by Fourier shell correlation (FSC) and line-profiles (Fig. 1b–d and Supplementary Figs. 7–8). Note that the actual resolution should be somewhat higher as the FSC reduces the signal-to-noise ratio of the data by a factor of two at all spatial frequencies. The oxidation states of the individual nano-plates were quantified from measurements of the OD at only two energies[13, 34–36] because, in three dimensions, thickness effects, which typically require measurement of off-resonant data for normalization can be neglected owing to the constant voxel size (see Methods). Thus, the chemical composition is computed directly from the polar angle in the correlation plot (Fig. 2a–b, see Methods). The distribution of compositions per voxel was clearly bimodal (centers of polar angle histogram: Li$_{0.93}$FePO$_4$ and Li$_{0.02}$FePO$_4$), in agreement with the bulk XRD measurement of the sample, yet the average state-of-charge (SOC) obtained by analyzing the 3D volume and independently measured 2D XAS data was 16.2% and 16.8%, respectively (Supplementary Figs. 10–11). The apparent discrepancy between this value and the 50% SOC of the electrochemical cell could occur if the population of particles harvested for the tomogram were located in a portion of the electrode with a deficiency in carbon content and/or electrolyte wetting, introducing transport deficiencies that delayed their reaction. The existence of two differentiated spectroscopic components is consistent with the presence of Li$_\alpha$FePO$_4$ ($\alpha \geq 0.9$) or Li$_\beta$FePO$_4$ ($\beta < 0.05$)[16, 27, 28]. Upon inspection of the resulting tomogram (Fig. 2c and Supplementary Movie 1), these components were found to coexist within the same particle in several cases.

**Chemical phase distributions of individual particles.** In the case of unlimited 3D spatial resolution and a binomial distribution, the upper limit of the composition error can be defined by the root-mean-square (RMS) widths of each Gaussian distribution of compositions (Fig. 2b and Methods), which are ±11.3% for Li$_\alpha$FePO$_4$ (Fe$^{2+}$-rich) and ±13.9% for Li$_\beta$FePO$_4$ (Fe$^{3+}$-rich). As a result, these measurements cannot resolve the small Li solubilities in the co-existing phases, revealed by XRD above, because they are smaller than the calculated error. Consequently, Li$_\alpha$FePO$_4$ is referred as LFP-rich, and Li$_\beta$FePO$_4$ as FP-rich. To reduce the impact of compositional error and enhance the resolution of domains that were chemically distinct, the voxels in the 3D map (Fig. 2c) were segmented into three major components, >70% Li$_\alpha$FePO$_4$ (LFP-rich), >70% Li$_\beta$FePO$_4$ (FP-rich), and mixed (i.e., 30–70% Li$_\alpha$FePO$_4$, the rest being Li$_\beta$FePO$_4$), with the

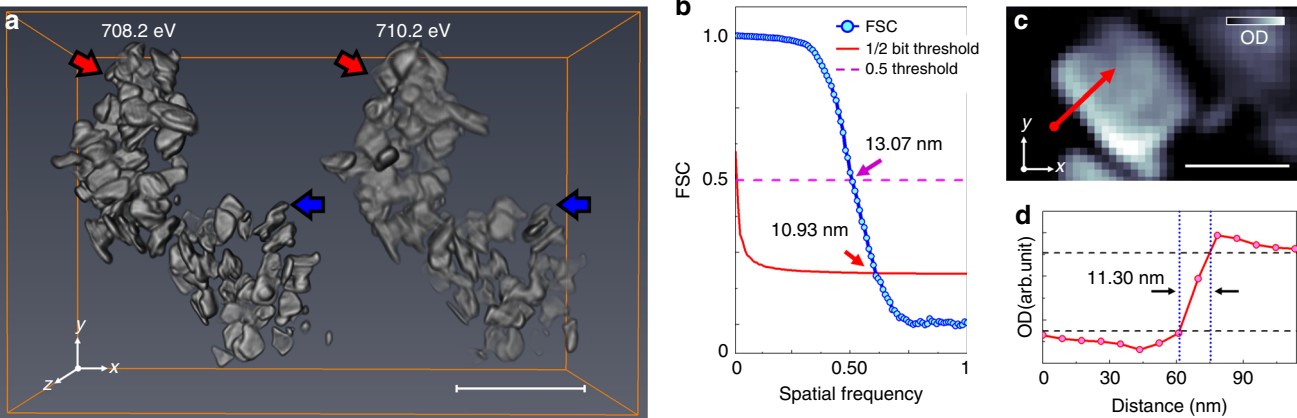

**Fig. 1** Results of tomographic reconstruction. **a** Reconstructed three-dimensional (3D) optical density volumes at 708.2 (left) and 710.2 eV (right). The size of reconstructed voxels is $6.7 \times 6.7 \times 6.7$ nm$^3$. **b** Resolution estimation of the 3D volume at 708.2 eV in **a** by Fourier shell correlation (FSC, blue solid line with scatter) with 1/2-bit (red solid line) and 0.5 (magenta dashed-line) threshold criteria. **c** Representative cross-section of the tomogram at 708.2 eV along the highest resolution plane ($xy$). The slice of the same position at 710.2 eV is shown in Supplementary Fig. 7. The positions of the slices are marked as red (cutting along $xy$ plane) and blue (cutting along $xz$ plane) arrows in **a**. The resultant cross-sections onto the lower resolution plane ($xz$ plane) at both 708.2 and 710.2 eV are shown in Supplementary Fig. 8. **d** Line profile indicated by the red arrow in **c**. Black-dashed lines are guides for 10–90% resolution criteria. Scale bars in **a** and **c** indicate 500 and 100 nm, respectvely

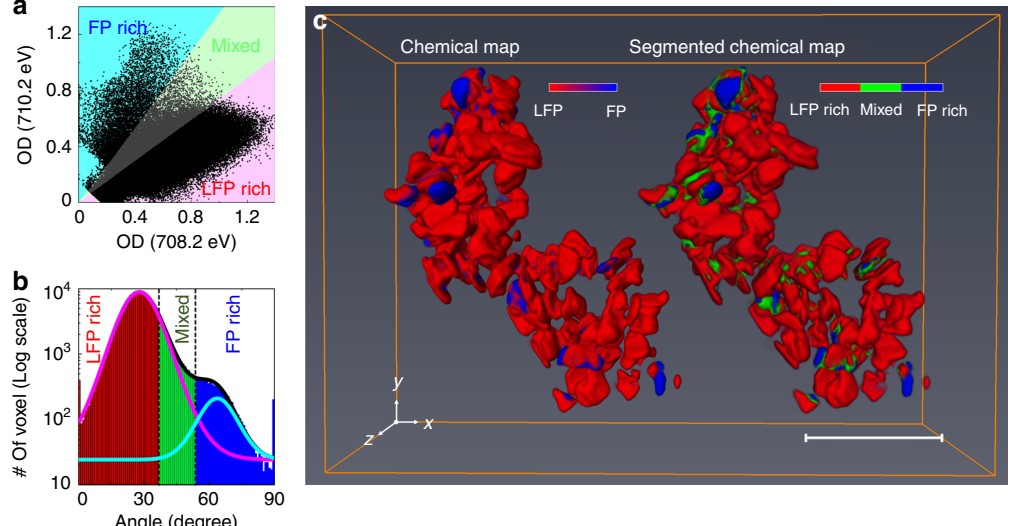

**Fig. 2** Three-dimensional (3D) chemical state mapping. **a** Correlative distribution plots between the optical densities (ODs) of each voxel at 708.2 and 710.2 eV. **b** Histogram plot of the polar angles of the data points in **a**. The $y$-axis is expressed as a logarithmic scale for better visibility. The plot can be fitted with summation (black solid line) of two Gaussian distributions which are centered on low (magenta solid line, 27.48°) and high (cyan solid line, 65.0°) polar angles correspond to Li$_{0.93}$FePO$_4$ and Li$_{0.02}$FePO$_4$, respectively. **c** 3D chemical map (left) and its segmentation into three chemical phase groups (right). The presence of the Li$_\alpha$FePO$_4$ (majority Fe$^{2+}$, LFP) and charged Li$_\beta$FePO$_4$ (majority Fe$^{3+}$, FP) were assigned colors red and blue, respectively (left). The voxels were separated into three distinct groups, indicating chemical phase group of each voxel, according to the polar angle. The red, green, and blue areas indicate LFP-rich (>70% Li$_\alpha$FePO$_4$), FP-rich (>70% Li$_\beta$FePO$_4$), and Mixed (30–70% Li$_\alpha$FePO$_4$, the rest being Li$_\beta$FePO$_4$) domains, respectively. The shading colors in **a**, **b** indicate the criteria for chemical segmentation. Scale bar, 500 nm

segmentation threshold at 30% (Supplementary Methods). It is important to emphasize that this segmentation purely reflects a conservative limit of detection of a given phase, and not its specific composition (e.g., $\alpha$ in Li$_\alpha$FePO$_4$). It gives a clear view of the most reliable information and is in agreement with a similarly segmented 2D XAS map with a total error of 7.4% (Supplementary Fig. 12). A total of 83 individual particles were segmented as shown in Fig. 3a. The fraction of particles with smaller dimensions in the tomogram was found to be higher in comparison with a larger, and, thus, more representative population of pristine particles imaged by transmission electron microscopy

(TEM, Supplementary Fig. 13). Though similarly shaped, the particles presented a variety of delithiation patterns, volumes, and total composition. A histogram of particle volumes and the corresponding fraction of activated particles is shown in Fig. 3b. The average composition of all voxels in each of the morphologically segmented particles was considered to determine particle activity. Active particles showed a statistically significant level of delithiation, defined as 15–100% Li$_\beta$FePO$_4$ based on a compositional error threshold (~13%). This activity threshold should both encompass our composition error and avoid misinterpretation of particles as active due to the solubility limit of Li$_\beta$FePO$_4$. While,

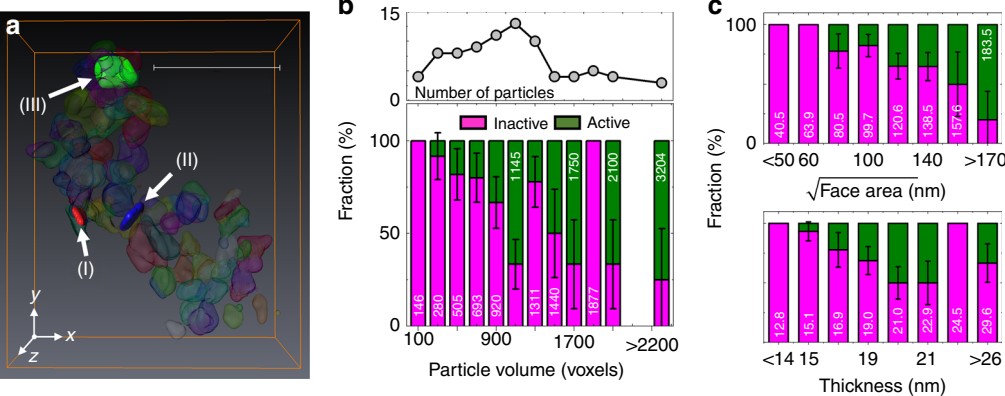

**Fig. 3** Activeness of each particle. **a** Voxel segmentation to define individual particles. Scale bar, 500 nm. **b** Volume distributions (black solid line with scatter) of individual particles shown in Fig. 2 and the fraction (bar plot) of inactive (magenta bar, <15% of Li$_\beta$FePO$_4$) and active (olive bar, 15–100% of Li$_\beta$FePO$_4$) particles as a function of particle volume with an increment of 200 voxels' volume. Averaged particle volume in each range also shown. Each voxel has a volume of 6.7 × 6.7 × 6.7 nm$^3$. The experimental error bars are calculated assuming a binomial distribution (active or inactive) taken at one standard deviation. **c** Compositional analysis based on the dimensions of each plate, comparing the facet area with thickness. The optical densities (ODs) of voxels along the particle thickness direction were averaged out across whole large facet. The thickness of the particle was calculated by the full-width-half-maxima of the averaged OD. The bar plots have the same color definition as **b**

overall, Li$_\alpha$FePO$_4$ was the major component across the range of particle volumes, 25 particles (30.1 ± 5.1 %) were defined as active if they had more than 15% Li$_\beta$FePO$_4$. Even with a tighter error threshold (25–100% Li$_\beta$FePO$_4$), 12.1 ± 3.6% particles are categorized as active. The error range is calculated assuming a binomial distribution (active or inactive) taken at one standard deviation. Only one of the (partly) delithiated particles was found to be close to the oxidized state (>70% Li$_\beta$FePO$_4$). A weak tendency was observed toward an increase in the population of active particles with volume (Fig. 3b). However, the variation in *ac* facet area (2500 ~40,000 nm$^2$) was much larger than particle thickness (10 ~30 nm). Tomography enabled the separate analysis of these particle dimensions, as shown in Fig. 3c. Interestingly, a significantly stronger dependence of the number of activated particles with facet size was found compared to overall volume. The averaged Li$_\alpha$FePO$_4$ concentrations for each individual particle showed a similar trend with respect to the facet area (Supplementary Fig. 14). Essentially no trend was found with particle thickness, likely due to the combination of a narrow distribution of values in this dimension and the small number of significant voxels along it due to finite spatial resolution. The fact that particles with small facets were both systematically found to be closer to the reduced state (Fig. 3c) and oversampled in the tomogram compared to the overall population of the electrode (Supplementary Fig. 13) could also explain the discrepancy between the overall SOC in the 3D image and the electrochemical cell.

Representative 3D chemical phase distributions for individual particles are shown in Fig. 4. These particles were chosen because their thickness was high enough compared to the spatial resolution to lead to insight into chemical gradients in all 3D. They also represent different degrees of delithiation. The particle showing the lowest degree of delithiation (Fig. 4a–c, overall composition Li$_{0.89}$FePO$_4$) showed evidence of reaction primarily around the edges (blue arrows) and along one of the two large facets (red arrows) of the crystal, without clear crystallographic directionality. The existence of multiple points of reaction propagation was confirmed in a second particle at a higher delithiation state (Fig. 4d–f, overall composition Li$_{0.81}$FePO$_4$). In this case, delithiation was also found to have occurred (red arrows) through the entire thickness of the particle. In contrast, a

sharp division between single, large lithiated, and delithiated domains was observed in a particle at a much higher state of delithiation (Fig. 4g–i, overall composition Li$_{0.41}$FePO$_4$).

## Discussion

The 3D snapshot of partly reacted states revealed the existence of sharp inhomogeneities within a particle, ascribed to co-existence of Li$_\alpha$FePO$_4$ or Li$_\beta$FePO$_4$, where $\alpha \gg \beta$ and $(\alpha-\beta)$ defines the miscibility gap. The exact values of $\alpha$ and $\beta$ could not be extracted at the chemical resolution of our measurement. Phase co-existence has been observed in nanoparticles by others[12, 18, 37, 38]. It has been shown to prevail during slow charging in recent operando X-ray studies[36, 39, 40]. This intra-particle heterogeneity, in ex-situ conditions, is inconsistent with a models of domino-cascade delithiation[19] or relaxation from a kinetic solid solution state through inter-particle exchange of Li[21, 41, 42], where populations of particles containing only either Li$_\alpha$FePO$_4$ or Li$_\beta$FePO$_4$ would be expected. The population of particles with detected heterogeneity is generally larger than in other studies[19, 34, 43, 44]. It is plausible that the higher spatial resolution achieved with ptychography improved the detection of heterogeneity. It is also worth noting that the average sizes in these previous studies were well above 100 nm, and, thus, much larger than here. Interestingly, Brunetti et al. reported that mixed particles were among the smallest within the studied population[45]. Particle-by-particle models of transformation[44] would not account for the large population of the particles with at least 15% of delithiated phase (i.e., active) observed in this study (30.1 ± 5.1 %). In contrast, this value was in good agreement with recent operando μ-XRD, where an average 22% active particles was reported during a charge at C/5[40], adding support to the relevance of the tomographic observations here to working conditions.

The distribution of delithiated domains in individual particles was complex and irregular irrespective of their individual Li content. Complex patterns of delithiation in LiFePO$_4$ nanoparticles have been recently predicted by Welland et al. using a thermodynamic 3D model, where the effect of coherency strain, elastic moduli, and surface wetting was considered[46]. Kinetically, such patterns could be aggravated by the existence of uneven electrical contacts within individual nanoparticles in a composite porous electrode, driven by local heterogeneities in component

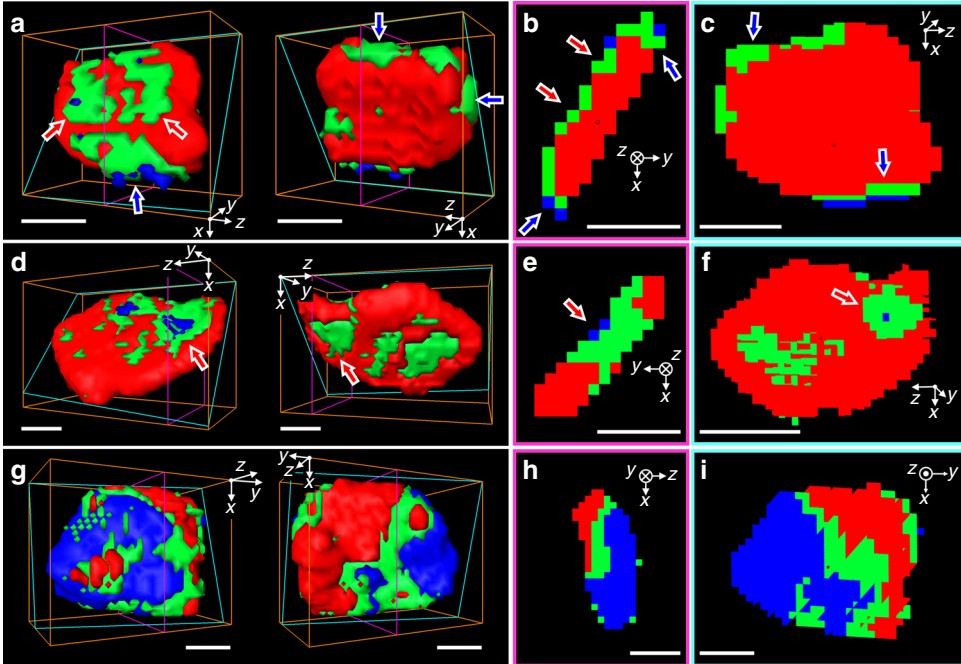

**Fig. 4** Representative three-dimensional (3D) chemical phase distribution of individual particle. **a**, **d**, **g**, Front (left) and backside (right) views of isosurface of three chemical components. Cross-sectional views along the thickness direction (**b**, **e**, **h**) and along the large face (**c**, **f**, **i**), respectively. The cross-section planes are indicated as magenta and cyan colored boxes in 3D isosurface plots. The red, green, and blue indicate LFP-rich, mixed, and FP-rich voxels, respectively. The positions of each particle are noted as (I), (II), and (III) in Fig. 3a for **a**, **d** and **g**, respectively. All scale bars, 50 nm

distribution. These contacts act as sinks of ions and electrons, and, thus, points of reaction initiation. If transient $Li_xFePO_4$ solid solutions were present under working conditions because phase separation is suppressed by coherency strain in these small particles[21, 47], a heterogeneous distribution of electrical contacts, and, thus, carrier diffusion, could impose gradients in $x$ within a particle. Such gradients were recently predicted to trigger immediate spinodal decomposition into two phases even under applied current[48], which would be consistent with our ex-situ observations. In a scenario where reaction progression is conditioned by electrical contacts, the correlation between large $ac$ facets and increased level of delithiation found over many particles, also suggested by others[46], could result from the larger probability of creating multiple points of electrical contact as facet area increases. Thermodynamic origins, imposed by coherency strain along the $ac$ facet, are also possible. Calculations of energy barriers to phase co-existence in a $Li_xFePO_4$ nanoparticle available in the literature are conflicting. The 3D model by Welland et al. led to an inverse relationship with size[46], which would predict larger nanoparticles reacting first, as observed here. In contrast, Cogswell et al. found lower or comparable barriers with decreasing size using a 2D model of the ac facet with depth averaging[49]. The discrepancy could, at least in part, be due to the different interfacial orientations in the two models, suggesting that further computational work would be helpful to establish this question.

Our demonstration of soft X-ray ptychographic tomography visualizes chemical states at a spatial resolution of 11 nm and is a powerful tool now available to chemists and materials scientists seeking insight into heterogenous chemical distributions occurring in 3D, even in nanocrystals within relatively large fields view. This feature provides the opportunity to analyze ensemble statistics. Resolving phenomena in 3D, at high resolution, allowed us to discern the effect of the dimensionality of anisotropies in transport and electrochemical reactivity. The combination with soft X-ray spectroscopic methods enables the analysis of chemical states in a variety of elements, from transition metals to common anions such as O. The resulting specimen thicknesses and required speeds are prohibitive for equivalent, high spatial resolution techniques in electron microscopy, making it an ideal complement to uncover phenomena at multiple scales in one measurement. This multiscale insight is critical to accurately defining the properties of functional materials in realistic architectures such as batteries.

## Methods

**Synthesis of $Li_xFePO_4$ (x ~0.5) nano-plates.** Plate-shaped LiFePO₄ were synthesized using a previously reported solvothermal method[12, 30]. H₃PO₄ and LiOH·H₂O were dissolved in ethylene glycol at 50 °C, followed by addition of FeSO₄·7H₂O, for a total 1:1.5:2.7 molar ratio. After stirring for 30 min, the mixture was heated at 180 °C for 10 h. In order to achieve good conductivity, LiFePO₄ nano-plates were carbon coated by mixing with 20 wt % of sucrose and then carbonizing at 650 °C for 3 h in Ar atmosphere[12, 30]. Powder X-ray diffraction patterns (Supplementary Fig. 3) were collected using a Bruker D8 Discover X-ray diffractometer operating with Cu $K\alpha$ radiation ($\lambda_{avg}$ = 1.5418 Å). They were consistent with mixtures of LiFePO₄ (JCPDS card number 40–1499) and FePO₄ (JCPDS card number 29–0715). In order to evaluate the macroscopic electrochemical properties of the oxide and prepare the specific state-of-charge (~50%), composite electrode films were fabricated by mixing the pristine oxide with acetylene black and polyvinylidene difluoride (PVDF) in a 80:10:10 ratio in N-methylpyrrolidone. The resulting slurry was cast onto a pre-weighed Al foil disk, dried at room temperature, followed by a heat treatment of 120 °C under vacuum. The composite electrodes were assembled in 2032 coin cells using lithium foil as both counter and pseudo-reference electrode, and Celgard 2400 separator soaked in a 45:55 mixture of ethylenecarbonate and dmiethyl carbonate containing 1 M LiPF₆ as electrolyte. All cell assembly and sample manipulation was performed in an Ar-filled glovebox. A Bio-Logic VMP3 potentiostat/galvanostat were used to carry out all electrochemical experiments in galvanostatic mode, at C/10 rate. Capacity at the first discharge of the preliminary cell are in good agreements with the literatures (156 mAh g⁻¹), showing that almost all particles were involved in the electrochemical reaction (Supplementary Fig. 1). The half-charged cell (~50% state-of-charge) for the imaging experiments was stopped at to 78 mAh g⁻¹ (Supplementary Fig. 1).

**Soft X-ray ptychographic microscope.** Soft X-ray ptychographic microscopy measurements were performed at the bending magnet beamline (5.3.2.1) at the Advanced Light Source (ALS), Lawrence Berkeley National Laboratory[11, 12]. Ptychographic measurements utilized a 100 nm outer zone width Fresnel zone plate for illumination and proceeded with a square scan grid of 70 nm steps. Diffraction patterns from 200 ms exposure were directly recorded on a custom fast readout CCD with the 5-μm-think $Si_3N_4$ attenuator to expand the dynamic range (Supplementary Fig. 4). The diffraction data were reconstructed by 500 iterations of an implementation of the RAAR algorithm[50]. Incoherent background noise was eliminated through the implementation of a background retrieval algorithm[11]. All data processing, including ptychographic reconstruction, and background retrieval were performed using standard methods available in the SHARP-CAMERA software package with parallel computation (http://camera.lbl.gov). The resolution of the individual 2D projection is calculated by Fourier ring correlation (FRC) to be 10 nm (½ bit threshold) at 708.2 eV (Supplementary Fig. 6).

**Registration of the rotation axis.** Aligning the 2D projections of a tomographic tilt series to a common rotation axis (not necessarily the real rotation axis) with sub-pixel resolution is essential to achieve a good quality 3D reconstruction. In order to achieve sub-pixel-precision, we have developed an iterative registration method with intensity-base automatic image alignments. To set the common rotation axis, the projections were first roughly aligned using an alignment feature only with translations of pixel size. The alignment features in all roughly aligned projections were close to the common rotation axis, but there still exist huge misalignments owing to inaccuracy of human interactions and tilting of each projection. We then reconstructed the 3D volume from the first aligned tomographic tilt series and computed 2D projections of the 3D volume according to the tomographic tilt angles. These computed 2D projections were used as reference images for second alignments. The second alignment was performed with intensity-based automatic image registration, which is an iterative process brings the misaligned image (2D projections of the tomographic tilt series) into alignment with the reference image (computed 2D projections). The process was performed following non-reflective similarity transformations (consisting of translation, rotation, and scale) to determine the specific-image transformation matrix that is applied to the moving image with bilinear interpolation. The same procedures were repeated until the aligned projections were self-consistent.

**Tomographic reconstruction.** Tomographic imaging proceeded from a series of 158 2D projections of the sample ODs recorded over a wide angular range from −80° to +77°. After the image registration, the OD volumes at 708.2 and 710.2 eV (Fig. 1a), with voxel size of $6.7 \times 6.7 \times 6.7$ nm$^3$, were reconstructed using the algebraic reconstruction technique (ART) with 20 iterations[51]. The 3D resolution is confirmed by FSC of the OD volume at 708.2 eV and indicates a 3D spatial resolution around 11 nm (see Fig. 1b and Supplementary Methods). A small improvement in the 3D resolution were observed by adopting different reconstruction algorithm with a large number of iterations (Supplementary Fig. 15), but the discrepancy did not affect the conclusions of the analysis. This value is confirmed by line-cuts through the volume, shown in Fig. 1c–d.

**Chemical phase quantification.** The OD volumes at 708.2 and 710.2 eV were used for estimating quantitative chemical information (e.g., the oxidation state of iron in $Li_xFePO_4$) at each voxel. From the standard spectra of the discharged $Li_\alpha FePO_4$ (majority $Fe^{2+}$) and charged $Li_\beta FePO_4$ (majority $Fe^{3+}$), the relative absorption intensity ($I_{E1,LFP}$, $I_{E1,FP}$, $I_{E2,LFP}$, and $I_{E2,FP}$) at specific energy ($E_1$ and $E_2$) were acquired (Supplementary Fig. 5). Since the amount of the absorption at a certain energy is linearly proportional to the relative amount of species with different iron oxidation states, the chemical concentration of $Li_\alpha FePO_4$ ($C_{LFP}$) and $Li_\beta FePO_4$ ($C_{FP}$) at each voxel can be calculated by the relation:

$$\begin{pmatrix} OD_{E1} - OD_{pre-edge} \\ OD_{E2} - OD_{pre-edge} \end{pmatrix} = \begin{pmatrix} I_{E1,LFP} & I_{E1,FP} \\ I_{E2,LFP} & I_{E2,FP} \end{pmatrix} \begin{pmatrix} C_{LFP} \\ C_{FP} \end{pmatrix} \qquad (1)$$

where $OD_{E1}$, $OD_{E2}$, and $OD_{pre-edge}$ indicate the single voxel OD at $E_1$, $E_2$, and pre-edge region, respectively. While normalization of all ODs with $OD_{pre-edge}$ can maximize the chemical contrast, because $OD_{pre-edge}$ is proportional to pure mass thickness without chemical contrast, the OD at pre-edge region was not clear enough to reconstruct 3D volume and negligible compared with ODs at 708.2 and 710.2 eV (Supplementary Fig. 9). Since the concentration of each chemical phase, $LiFePO_4$ ($Fe^{2+}$) and charged $FePO_4$ ($Fe^{3+}$), can be expressed as linear equations corresponding to the OD volumes at 708.2 and 710.2 eV, the polar angle in the correlation plot is a function of the relative compositions of two major elements for the corresponding voxel (Fig. 2b). As a result, the 3D distribution of Fe oxidation state can be retrieved quantitatively (Fig. 2c). The fidelity of the 3D chemical map obtained in this way is verified by projecting the calculated volume along the z-axis and comparing with a map obtained by a linear combination fit of the reference spectra to independently measured 2D XAS data across the full spectrum of the same sample (Supplementary Figs. 11–12). The accuracy of the chemical map from 2D XAS data is represented by R-factor, which is less than 0.15 in 94.64% of pixels (Supplementary Fig. 10).

**Data availability**. The data that support the findings of this study are available from the corresponding author (D.A.S. or J.C.) on request.

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

## Acknowledgements

Soft X-ray ptychographic microscopy was carried out at beamline 5.3.2.1 at the Advanced Light Source. The Advanced Light Source is supported by the Director, Office of Science, Office of Basic Energy Sciences, of the U.S. Department of Energy under Contract No. DE-AC02-05CH11231. This work was supported as part of the NorthEast Center for Chemical Energy Storage, an Energy Frontier Research Center funded by the U.S. Department of Energy, Office of Science, Office of Basic Energy Sciences under Award Number DE-SC0012583. C.K. acknowledges additional support by the National Research Lab (NRF-2015R1A2A1A01006192) program of the National Research Foundation of Korea. This work is partially supported by the Center for Applied Mathematics for Energy Research Applications (CAMERA), which is a partnership between Basic Energy Sciences (BES) and Advanced Scientific Computing Research (ASRC) at the U.S Department of Energy.

## Author contributions

Y.-S.Y., C.K., J.C., and D.A.S. conceived of and planned the experiment. D.A.S., T.T., R.C., P.D., J.J., H.K., F.R.N.C.M., A.L.D.K., T.P.C.L., T.W., Y.-S.Y., and H.P. developed experimental techniques, software, and equipment. D.A.S. and S.M. developed ptychographic reconstruction codes. Y.-S.Y., C.K., F.C.S., and C.P.G. prepared the samples. Y.-S.Y., M.F., and D.A.S. carried out the ptychographic microscopy measurements. Y.-S.Y., Y.L., and D.A.S. performed post-experiment data analysis, and Y.-S.Y., Y.L., C.K., D.A.S., and J.C. established the interpretation of the chemical maps. Y.-S.Y., C.K., J.C., and D.A.S. prepared the manuscript, which incorporates critical input from all authors.

## Additional information

**Competing interests:** The authors declare no competing interests.

