## [Peer Review File · Nature Communications]

Reviewers' comments:

Reviewer #1 (Remarks to the Author):

This article deals with the three dimensional localization of nanoscale battery reactions using soft X-Ray tomography. Over all the paper is well written. But some points need to be clarified and the manuscript needs to be revised.

Nevertheless, in the introduction, the authors have to develop electron-based technique (add recent references) and should also be considered 3D EFTEM tomography which is a powerful technique to precisely quantify the chemical state of material. The 3D EFTEM tomography literature should also be added by the authors.

The precise chemical localization of LiFePO₄ (LFP) and FePO₄ (FP) have already been studied by XRD and by the combination of TEM/EELS techniques teen years ago. This study have inspired the authors for the XAS fit of Fe-L3 and L2 edges. This study have to be mentioned on XAS results and have to compare this result in the discussion part.

The authors claims that the nano-plates were electrochemically delithiated until 50% of the total amount of lithium but no precise chemical quantification using XRD have been done in order to clearly determine the x value Li_xFePO₄. This fact have to be determined in order to precisely quantify the chemical composition by XAS. The authors have to explain why the chemical composition is not precisely determined by this technique.

XAS results: Fig S9 c): the fit result by combination of LFP and FP do not exactly match with XAS experimental data. The intensity of Fe-L3 edge is different. The authors have to discuss this difference in the paper.

Fig S10: 2D or 3D chemical maps do not fit with the Li_{0.5}FePO₄ composition of the nano-platelet. The authors have to discuss about this fact.

Fig 4b,e,h: cross-section views show pixel-based reconstruction due to the limited resolution of 20 nm. This resolution is not sufficient to clearly study the 3D chemical map. This fact should also be considered by the authors.

The OD volumes were reconstructed using ART technique given pixel-based images for cross-section views. The reconstruction could be improve using SIRT technique. Did the authors try this technique of reconstruction?

No specific delithiated mechanism was proposed by the authors from this study; the authors compared their results to mechanisms proposed by other authors but their results cannot be explained probably due to a problem with the electrochemical delithiation.

Reviewer #2 (Remarks to the Author):

The authors report on the development of X-ray tomography allowing to resolve chemical state with a 3D 11 nm resolution, which is applied to nano plates of partially oxidized LiFePO₄. In my opinion this technique is demonstrated to be very powerful in determining distributions in chemical states, thereby locating phase boundaries, which is very relevant for battery materials.

The technical description of the technique appears clear and convincing, however in my opinion the scientific discussion of the results is missing a number of important components. In this context I have the following comments:

Line 66: The authors mention observed metastable pathways in LFP and then refer to references. [14, 16-20]. However, 14, 16-18 do not observe metastable pathways (16 even states it explicitly does not see solid solutions and 17 performs powder XRD observing only LFP and FP and suggests the domino

cascade model based on the absence of XRD broadening). Ref 19 is a theoretical study. Only ref 20 observes meta stable phases indeed, so do Orikasa et al [dx.doi.org/10.1021/ja312527x](https://doi.org/10.1021/ja312527x) and Zhang et al [dx.doi.org/10.1021/nl404285y](https://doi.org/10.1021/nl404285y), which are not referred to.

Line 143: The authors mention that these results can be used to validate existing models. I am missing comparison to phase field modeling results in ref 24 which appear as one of the most relevant and important modeling efforts on LFP in literature.

Line 158. It would be interesting and highly relevant to compare the active particle fraction with references 24 and 30, where it is specifically reported and discussed.

It is established that the particle size both influences the open cell potential and the nucleation kinetics, is it possible that this plays a role in these experiments? Did the authors delithiate with different charge rates? This would be a very relevant and interesting parameter to vary.

I suggest provide the reader with information to what extend this technique can be applied. For instance are there limitations towards the edge energies that can be probed (limiting it to certain elements for example). Is operando a possibility?

Three-dimensional localization of battery reactions using soft X-ray tomography

Authors' Response

We thank the editors and the referees for the careful review of the manuscript. Based on the reviewer's comments, we made significant modifications to the manuscript and present new results. In this document, the reviewer comments are *italics*. The changes to the text are in **bolded**.

Reviewer #1

Comment #1:

"This article deals with the three dimensional localization of nanoscale battery reactions using soft X-ray tomography. Over all the paper is well written. But some points need to be clarified and the manuscript needs to be revised."

Reply #1:

We sincerely appreciate the greatly positive judgment the reviewer does of this manuscript. We have made all of the suggested changes, and believe that they have significantly improved the manuscript.

Comment #2:

"Nevertheless, in the introduction, the authors have to develop electron-based technique (add recent references) and should also be considered 3D c tomography which is a powerful

technique to precisely quantify the chemical state of material. The 3D EFTEM tomography literature should also be added by the authors.”

Reply #2:

We agree that EFTEM tomography is a very powerful method for analyzing the chemical states of nano-materials in three dimensions. However, its application is limited to samples with small dimensions, because of limitations in object thickness at any given rotation. Particularly when studying crystalline materials, as we do here, elemental maps also suffer from the presence of crystal-lattice diffraction contrast, to which we are not sensitive. Furthermore, jump ratio maps are not quantitative for materials that are much thicker than the mean-free-path for inelastic electron scattering. Our sample is nearly 1 micrometer thick along its thickest dimension. We do not believe it would be possible to image such a material quantitatively with TEM. We have added the following statements (highlighted sentences in page 3, lines 10~13) and added a new reference (current ref. 10),

In turn, while electron-based techniques achieve very high spatial resolution⁷⁻⁹ and can provide three-dimensional (3D) quantification of the chemical state¹⁰, they also suffer from diffraction contrast effects and non-linearities for material thicknesses greater than the mean-free-path of inelastic scattering.

Comment #3:

“The precise chemical localization of LiFePO₄ (LFP) and FePO₄ (FP) have already been studied by XRD and by the combination of TEM/EELS techniques teen years ago. This study have inspired the authors for the XAS fit of Fe-L3 and L2 edges. This study have to be mentioned on XAS results and have to compare this result in the discussion part.”

Reply #3:

We certainly do not claim to be the first researchers to localize chemical states in $\text{LiFePO}_4/\text{FePO}_4$ by fitting the Fe- $L_{2/3}$ spectra [Shapiro, D. A. *et al.*, *Nature Photon.* **8**, 765 (2014); Yu, Y.-S. *et al.*, *Nano Lett.* **15**, 4282 (2015); Li, Y. *et al.*, *Adv. Funct. Mater.* **25**, 3677 (2015); Lim, J. *et al.*, *Science* **353**, 566 (2016); Chueh, W. C. *et al.*, *Nano Lett.* **13**, 866 (2013); Li, Y. *et al.*, *Nature Mater.* **13**, 1149 (2014)]. We have added references to representative EELS, soft XAS and XRD reports in the analysis (pages 4 and 5) and discussion (page 7) sections, which have been expanded to fully address the concern of the reviewer. However, we note that, unlike TEM-EELS, the XRD reports available do not provide chemical localization within nanoparticles, as the reviewer seems to imply. In this context, we again wish to note that the benefit of soft X-ray scanning microscopy (STXM) is that our spatial resolution is significantly higher than in the XRD reports in the LiFePO_4 literature (though we are not sensitive to atomic positions) and soft X-rays can penetrate more material than TEM. Thus, STXM combined with XAS is highly complementary to XRD and TEM/EELS. It is also important to highlight that none of these studies report the quantification of chemical states in 3D, as we do here.

Comment #4:

“The authors claims that the nano-plates were electrochemically delithiated until 50% of the total amount of lithium but no precise chemical quantification using XRD have been done in order to clearly determine the x value Li_xFePO_4 . This fact have to be determined in order to precisely quantify the chemical composition by XAS. The authors have to explain why the chemical composition is not precisely determined by this technique.”

“Fig S10: 2D or 3D chemical maps do not fit with the $\text{Li}_{0.5}\text{FePO}_4$ composition of the nano-platelet. The authors have to discuss about this fact.”

Reply #4:

We wish to clarify that electrochemistry is indeed a precise way to measure chemical composition when the electrodes are operated at the redox potentials of LiFePO_4 , below 4 V vs. Li^+/Li^0 , because essentially no side reactions occur [Castro, L. *et al.*, *J. Electrochem. Soc.* **159**, A357 (2012)]. In fact, it is widely accepted that electrochemical analysis techniques can be among the most sensitive available. Nonetheless, we have measured bulk XRD data for our partly delithiated powder, and compared it with the pristine state, LiFePO_4 , as well as the fully delithiated state, FePO_4 . The data has been added to the manuscript as Fig. S3. Comparison of the patterns with analysis of similar XRD data abundant in the literature [Hess, M. *et al.*, *Nat. Commun.* **6**, 8169 (2015); Yamada, A. *et al.*, *Nature Mater.* **5**, 357-360 (2006); Wagemaker, M. *et al.*, *J. Am. Chem. Soc.* **133**, 10222-10228 (2011)] confirmed that the sample indeed contains an approximate 50% ratio of $\text{Li}_\alpha\text{FePO}_4$ and $\text{Li}_\beta\text{FePO}_4$, where α and β were slightly smaller than 1 and very close to 0, respectively. Therefore, this result is in line with the coulometric quantification from the Li metal half-cell. However, this quantity DOES NOT pertain to any single nanoplate, but rather reflects the average over the entire electrode used in the coin cell. Individual nanoplates, and even large domains of particles [Liu, J. *et al.*, *J. Phys. Chem. Lett.* **1**, 2120 (2010)], can have a chemical state that differs from the average of the electrode, as well as among themselves. Such discrepancies have been reported before [Boesenberg, U. *et al.*, *Chem. Mater.* **25**, 1664 (2013); Chueh, W. C. *et al.*, *Nano Lett.* **13**, 866 (2013); Li, Y. *et al.*, *Adv. Funct. Mater.* **25**, 3677 (2015); Li, Y. *et al.*, *Nature Mater.* **13**, 1149 (2014); Zhang, X. *et al.*, *Nat. Commun.* **6**, 8333 (2015)], and are within reason for the very nature of a heterogeneous, two-phase, chemical phase transformation.

We also wish to emphasize that soft X-ray absorption spectroscopy DOES precisely quantify the Fe valence state. Further, if a heterogeneous mix of pure states (i.e. Fe^{2+} and Fe^{3+})

exists, STXM, combined with X-ray absorption spectroscopy precisely quantifies the spatial distributions of the various states. For a complete discussion see current refs 11,12,34,36,42,43 [Shapiro, D. A. *et al.*, *Nature Photon.* **8**, 765 (2014); Yu, Y.-S. *et al.*, *Nano Lett.* **15**, 4282 (2015); Li, Y. *et al.*, *Adv. Funct. Mater.* **25**, 3677 (2015); Lim, J. *et al.*, *Science* **353**, 566 (2016); Chueh, W. C. *et al.*, *Nano Lett.* **13**, 866 (2013); Li, Y. *et al.*, *Nature Mater.* **13**, 1149 (2014)]. Our measurements clearly show the existence of two distinct Fe chemical phases, consistent with Fe²⁺ and Fe³⁺. In parallel, our XRD results indeed indicate that the sample is a heterogeneous mixture of LiFePO₄ and FePO₄, supporting the assignment of spectroscopic signals.

Under these circumstances, our X-ray absorption spectroscopy results did precisely quantify the chemical composition at each point within the sample and the average level of delithiation of the particles in the tomogram is simply determined by counting the voxels of each component. These experiments were carried out on a small portion of the electrode at 50% SOC. Selection of the specific portion was largely random because no methods exist to pick particles that reflect the exact distribution of compositions of the entire electrode. Therefore, since the transformation is heterogeneous, it is not at all surprising that a small, random sampling of the total battery could have an overall composition that is different from the entire bulk, just as there are many voxels within our tomogram that are either fully charged or fully discharged. As already explained in the manuscript, we found that the portion we measured had a larger ratio of particles with small facets than the overall population of the electrode, by comparing the histogram of particle sizes from TEM to the histogram of particles analyzed. The trend for particles with smaller facets to be less delithiated revealed by our ptycho-tomography analysis provides a robust explanation to the lower overall delithiation level in the tomogram compared to the overall electrode, as already discussed in pages 6 and 7. In other words, the discrepancy not only is not a sign of issues with the

determination of composition by XAS, but can also be explained with the unique insight revealed by chemically resolved ptychographic tomography, a key advance demonstrated by this paper.

Comment #5:

“XAS results: Fig S9 c): the fit result by combination of LFP and FP do not exactly match with XAS experimental data. The intensity of Fe-L3 edge is different. The authors have to discuss this difference in the paper.”

Reply #5:

As with any experimental measurement, there are finite uncertainties in the compositions we calculate which originate from statistical fluctuations in our data. The XAS in the original Fig. S9c (in revised version, Fig. S10c) are extracted from single 6.67 nanometer pixels, marked as the yellow dots in Fig. S9a (in revised version, Fig. S10a). This area is the smallest ever probed by an X-ray microscopy and, owing to such small sampled volumes, the signal-to-noise ratio of the single pixel spectra is low, which provides a straightforward explanation to why the fitting accuracy should be worse than for spectra collected by bulk techniques. To overcome the limitation, pixels showing unacceptable signal-to-noise ratios were filtered out, which are primarily located outside the object. The quality of each LC fit for each single pixel is also confirmed with an *R*-factor map (original Fig. S9b, in revised version, Fig. S10b) defined as, $R = \sum(\text{data-fit})^2 / \sum(\text{data})^2$. 94.64% of pixels were fit with *R*-factors of less than 0.15. For comparison, we added a new supplementary figure (Fig. S10c) which shows the XAS extracted from the entire object area and its LC fits. As the reviewer will notice, adding up the data for many pixels reduces the experimental error in the spectrum, and results in significant improvements of the fit,

reinforcing our explanation. We wish to emphasize that we analyzed the error in our measurement of composition extensively by multiple methods. In addition to the R -factor used for the two dimensional projection map, we analyzed the error in the three-dimensional composition map using the relative widths of a bi-modal distribution. For the case of infinite spatial resolution and a perfectly two-phase composition the distribution of composition values would be narrowly peaked. The widths we measure, ~15% RMS, therefore represent an upper bound on compositional uncertainty and is in agreement with the R -factor values.

Comment #6:

“Fig 4b,e,h: cross-section views show pixel-based reconstruction due to the limited resolution of 20 nm. This resolution is not sufficient to clearly study the 3D chemical map. This fact should also be considered by the authors.”

Reply #6:

The three dimensional spatial resolution of our tomograms is 11 nm, not 20, as the reviewer claims. To confirm the spatial resolution of the 3D volume, we have adopted the 3D Fourier shell correlation, which can calculate normalized cross-correlation coefficients between two independent 3D volumes from the identical object. In order to get two independent 3D volumes, the 158 OD projections at each X-ray energy were divided into two separate subsets corresponding to even and odd tomographic tilt angles (θ), and then were reconstructed as an independent OD volume of the same object. Thus, the actual resolution from the entire tilt series should be somewhat higher as the FRC reduces the signal-to-noise ratio of the data by a factor of two. To emphasize this fact, we added the following statement (highlighted sentences in page 5).

Note that the actual resolution should be somewhat higher as the FSC reduces the signal-to-noise ratio of the data by a factor of two at all spatial frequencies.

We also concur that the sampling of information along the particle thickness is limited since the thickness is only somewhat larger than our spatial resolution. We have edited a sentence in page 7 to read:

Essentially no trend was found with particle thickness, likely due to the combination of a narrow distribution of values in this dimension and the small number of significant voxels along it because of finite spatial resolution.

This shared concern also led us to analyze the tomogram for individual particles (Fig. 4), only when those particles were significantly thick compared to the spatial resolution, as stated in page 8. This fact meant that multiple pixels were resolved along the thickness of the plate, making the analysis meaningful.

Comment #7:

“The OD volumes were reconstructed using ART technique given pixel-based images for cross-section views. The reconstruction could be improve using SIRT technique. Did the authors try this technique of reconstruction?”

Reply #7:

We thank reviewer 1 for bringing up this inquiry. ART [Algebraic Reconstruction Techniques; Gordon *et al.*, *J. theor. Biol.* **29**, 471 (1970)] and SIRT [Simultaneous Iterative Reconstruction Technique; Gilbert, *J. theor. Biol.* **36**, 105 (1972)] perform an iterative

algorithm for reconstruction so that the calculated projections of the reconstructed volume are compared with the experimental projections obtained by the microscope to maximize the similarity of the reconstructed data. Since the modifications of the reconstructed volume are based on the averaging over several projections, SIRT has generally generated smoother reconstructions, at a cost of slower convergence. For a direct comparison, we have performed three different types of reconstruction (ART – 20 iterations, SIRT – 20 iterations, and SIRT – 100 iterations) for the same data set and calculated Fourier Shell Correlation (FSC) in order to confirm the quality of the reconstructions. For SIRT, a small improvement in the 3D resolution were observed only when we used a large number of iterations owing to its slow convergence. The discrepancy of the spatial resolutions from SIRT (10.42 nm with half bit threshold) and ART (10.93 nm with half bit threshold) does not affect the main conclusions of this manuscript. A new supplementary figure (Fig. S14) and explanatory text in the Methods section have been added to clarify the effect of the reconstruction methods.

After the image registration (see Supporting Information), the OD volumes at 708.2 and 710.2 eV (Fig. 1a), with voxel size of $6.7 \times 6.7 \times 6.7 \text{ nm}^3$, were reconstructed using the algebraic reconstruction technique (ART) with 20 iterations⁴⁹. The 3D resolution is confirmed by Fourier shell correlation (FSC) of the OD volume at 708.2 eV and indicates a 3D spatial resolution around 11 nm (see Fig. 1b and Supporting Information). A small improvement in the 3D resolution were observed by adopting different reconstruction algorithm with a large number of iterations, but the discrepancy did not affect the conclusions of the analysis (see Fig. S14).

Comment #8:

“No specific delithiated mechanism was proposed by the authors from this study; the authors compared their results to mechanisms proposed by other authors but their results cannot be explained probably due to a problem with the electrochemical delithiation.”

Reply #8:

From the measured electrochemistry (Fig. S1) and XRD (Fig. S3) of the electrochemically prepared, partly delithiated nanoplates, we cannot detect any problems, and the reviewer provides no further details that we can address. We have provided a robust explanation above to the only possible discrepancy we could identify, namely the comparison between the delithiation levels of the sample analyzed and the entire electrode. Because our analysis offers a snapshot of the chemical states at one point of the electrochemical reaction, extensive comments on the precise mechanism, which require dynamic observations, are not warranted. Nonetheless, we believe we have demonstrated that a significant technical advance also brought about novel insight into the distribution of reacted domains upon delithiation. More specifically, our ability to resolve the chemical states within the particles in three dimensions (3D) with unprecedented spatial resolution has clearly revealed intraparticle heterogeneity and multiple reaction points within single particles, without having to infer the existence of individual particles, as it is done in two dimensional (2D) measurements. The complete dimensionality of the domain boundaries was uncovered, which revealed differences in the correlation of particle size along different directions and extent of delithiation. No other analytical technique currently exists which can provide this information.

The mechanisms of delithiation of LiFePO_4 have been the object of extensive theoretical and experimental studies, with a variety of models available in the literature. In this context, it is natural for us to simply compare our results to existing models as opposed to “reinvent the wheel” by coming up with a new one. This comparison was made in order to

provide validation through the lens of 3D observations that were not experimentally available before. Should no models have been able to explain our observations, we would indeed have needed to formulate a new one. This was not the case, as argued in the manuscript. By comparison of our observations with predictions of *ex-situ* states by different models in the literature, we conclude that the patterns of heterogeneity and, indirectly, the dependence of degree of deintercalation with plate facet likely result from non-uniform connectivity between the particles and the electrode architecture. This specific effect we identify was referred to simply as “surface wetting” in available models, and it provides a satisfactory explanation. This insight has also not been revealed by any other study to date.

Reviewer #2

Comment #1:

“The authors report on the development of X-ray tomography allowing to resolve chemical state with a 3D 11 nm resolution, which is applied to nano plates of partially oxidized LiFePO₄. In my opinion this technique is demonstrated to be very powerful in determining distributions in chemical states, thereby locating phase boundaries, which is very relevant for battery materials.”

Reply #1:

We thank reviewer 2 for emphasizing that our X-ray ptycho-tomography is unique, which can bring the new insights relevant for battery technology. Our work is a collaboration which aims to develop unique and powerful microscopy tools which will have fundamental impact on the materials sciences.

Comment #2:

“The technical description of the technique appears clear and convincing, however in my opinion the scientific discussion of the results is missing a number of important components. In this context I have the following comments:”

Reply #2:

We sincerely appreciate the greatly positive judgment the reviewer does of this manuscript. We have made all of the suggested changes, and believe that they have significantly improved the manuscript.

Comment #3:

“Line 66: The authors mention observed metastable pathways in LFP and then refer to references. [14, 16-20]. However, 14, 16-18 do not observe metastable pathways (16 even states it explicitly does not see solid solutions and 17 performs powder XRD observing only LFP and FP and suggests the domino cascade model based on the absence of XRD broadening). Ref 19 is a theoretical study. Only ref 20 observes meta stable phases indeed, so do Orikasa et al [dx.doi.org/10.1021/ja312527x](https://doi.org/10.1021/ja312527x) and Zhang et al [dx.doi.org/10.1021/nl404285y](https://doi.org/10.1021/nl404285y), which are not referred to.”

Reply #3:

We have reviewed and added primary references. The positions of the references were incorrect, introduced due to a compilation error, and understandably caused confusion. We have corrected error (highlighted sentences in page 4) and added the reference (refs 23,24) in order to fulfill the reviewer’s request.

The mechanism of transformation of LiFePO_4 is one of the most intensely studied reactions in battery chemistry. While the reaction proceeds through a first order transition in equilibrium^{16,18-20}, under certain kinetic conditions, metastable pathways based on solid solutions have been observed²¹⁻²⁴.

Comment #4:

“Line 143: The authors mention that these results can be used to validate existing models. I am missing comparison to phase field modeling results in ref 24 which appear as one of the most relevant and important modeling efforts on LFP in literature.”

“Line 158. It would be interesting and highly relevant to compare the active particle fraction with references 24 and 30, where it is specifically reported and discussed.”

Reply #4:

We have grouped these two comments made by the reviewer at different points of their assessment because they are related. Among our particle sizes (50 ~ 170 nm along longest dimension) and with such a slow charging rate (C/10) it is predicted that the Li intercalation goes through a particle-by-particle transformation, according to the phase-field model the reviewer mentions [Li, Y. *et al.*, *Nature Mater.* **13**, 1149-1156 (2014)]. However, the population of the actively transforming particles (30.1 ± 5.1 %) is much higher than expected by the particle-by-particle transformations. These values are in good agreement with recent experimental result using *operando* μ -XRD that observed an average 22% active particles during charging at C/5, respectively [Zhang, X. *et al.*, *Nat. Commun.* **6**, 8333 (2015)]. To emphasize these facts, the following statements have been added in the manuscript (highlighted sentences in page 9)

Beyond these specific observations, the particle-by-particle model of transformation proposed to apply when the charging rate is slow (e.g., C/10 here) by phase-field simulations⁴³ would not account for the large population of the particles with at least 15% of delithiated phase (i.e., active) observed in this study (30.1 ± 5.1 %). In contrast, the fraction of such active particles was found to be in good agreement with the observation from a recent *operando* μ -XRD, which was on average 22% during a charge at C/5 (ref. 40), adding support to the relevance of the tomographic observations here to working conditions.

Comment #5:

“It is established that the particle size both influences the open cell potential and the nucleation kinetics, is it possible that this plays a role in these experiments? Did the authors delithiate with different charge rates? This would be a very relevant and interesting parameter to vary.”

Reply #5:

We agree that this important point is worth a great deal of study. A few members of our group participated in an *operando* study, which quantified the heterogeneity of the transformation as a function of charging rate. These measurements tracked multiple particles in 2D projection over the full charge cycle for multiple charging rates. The results of that study are relevant to the current work and are cited as a current ref. 36 [Lim, J. *et al.*, *Science* **353**, 566 (2016)]. In the early stages of our technical development reported here, throughput is limited so it has been prohibitive to try to record multiple tomograms from various charging rates. We hope that with future work our speed will be greatly increased so we can observe changes while tuning various experimental parameters.

Comment #6:

“I suggest provide the reader with information to what extend this technique can be applied. For instance are there limitations towards the edge energies that can be probed (limiting it to certain elements for example). Is operando a possibility?”

Reply #6:

We thank the reviewer for another interesting suggestion. Regarding *operando*, please see our response to Comment #5 above. In general, soft X-ray synchrotron sources can provide radiation sensitive to the *K*-edges of light elements, the *L*-edges of the transition metals and the *M*-edges of heavier elements. The transition metals are obviously very important to many areas of materials science, particularly with regard to batteries, so soft X-rays are a very powerful analytical probe. As with electrons, however, material thickness is a concern. We cannot achieve the spatial resolution of an electron microscope but we can penetrate through nearly 10 times more material. We now note this in the introduction where we refer to energy filtered TEM, as well as in the last paragraph of the manuscript. We feel that the combination of chemical sensitivity and material penetration will make our methods highly valuable to material researchers.

Reviewers' comments:

Reviewer #2 (Remarks to the Author):

The authors have convincingly revised the manuscript following the suggestions of all reviewers. Therefore I recommend publication without further revisions.

Reviewer #3 (Remarks to the Author):

This manuscript describes the development of soft X-ray ptychographic tomography and its application to the characterization of phase distribution in partially delithiated LiFePO₄ nanoparticles. The combination of 3D chemical information, a higher spatial resolution than hard x-ray tomography and a larger field of view than electron microscopy makes it a unique and powerful technique, which can potentially provide valuable insights complementary to other existing techniques especially in revealing the physics of battery materials. This work represents a notable progress in method development that in my opinion is potentially worthy of publication in Nature Communications.

My concern about the manuscript mainly lies in the interpretation and discussion of the observations made with this technique. The authors present an intriguing finding, namely a trend of increasing active particle population with (010) particle surface area (Fig. 4c) and particle volume (Fig. 4b). They attribute this phenomenon to non-uniform electrical contact between particles and the conductives. It's stated that "the correlation between large ac facets and increased level of delithiation... would be a trivial result of the larger probability of creating multiple points of electrical contact as facet area increases." I doubt the results presented are sufficient to support this conclusion. The analysis and discussion should be revised and strengthened by taking other factors into consideration. Below are several comments and questions I'd like to ask the authors to address.

1. I'm not fully convinced that the geometric argument made by the authors is the only explanation for the observed size-dependent degree of delithiation. As a counter argument, I want to bring to the authors' attention another possible explanation based on the competition between coherent vs incoherent nucleation. Several recent studies on the hydrogenation of Pd nanoparticles show that coherency stress is responsible for a similar size-dependent phase transformation behavior upon H (de)insertion (see Ulvestad et al. Nature Mater 2017, 16, 565 and references therein). Below a critical size, Pd particles can sustain insertion-induced stress without nucleating defects. The presence of stress increases the nucleation energy barrier and requires a larger driving force (H₂ gas pressure for Pd-H system) to initiate phase transformation. In contrast, particles above the critical size are able to form defects to relax stress, and stress reduction allows phase transformation to proceed at a lower driving force. Can a similar mechanism operate in LiFePO₄, so that a low driving force (overpotential) at the slow C/10 rate can trigger transformation of larger, defect-forming particles but is not sufficient for initiating nucleation in smaller coherent particles? This indeed seems to be consistent with the authors' earlier work in Ref. 12 and 11 that shows the presence/absence of cracks in large/small LiFePO₄ particles, although the cracked microcrystals studied there are much bigger than the nanoplatelets in this work. It is desirable that the authors can present additional results to prove the correlation or non-correlation between the degrees of delithiation and coherency stress, but at least please comment on this possibility.

2. I find the large active particle population seen here compared to other systems (e.g. Chueh's work) is consistent with the presence of large stress in the small nanoplates as coherency stress can cause the energy barrier for phase growth to increase with SOC and thus prevent the particle from complete transformation. Therefore a discussion on the potential effect of stress appears to be necessary.

3. Despite the statement "The apparent discrepancy between this value and the 50% SOC of the electrochemical cell is addressed below" on page 5, the authors do not seem to address this point head-on in the manuscript. A more complete discussion is needed. Reading between the lines, I think the authors suggest the reason for the discrepancy also lies in the dependence of electrical contact area on particle size. However, to support this claim, a plot of particle SOC vs the (010) facet area will be useful to demonstrate the trend instead of the distribution of active particle fraction shown in Fig. 3. In addition, can the authors exclude the possibility that the local electrode region where the cluster of nanoplate particles were imaged happened to have low carbon content and be poorly electronically wired?

4. On page 9, the authors state that "The existence of multiple points of reaction initiation within a particle can only be explained by a recently proposed model where surface wetting was considered in addition to coherency strain and elastic moduli within nanoparticles of similar dimensions." I question the need and value of invoking the "surface wetting" concept here. The local initiation of reaction on surface could simply be understood as heterogeneous nucleation events, which is the first step in the nucleation and growth process. Even if the particle surface has a uniform electronic contact with carbon and electrolyte network, reaction will still happen locally on surface (i.e. nucleation) at a low driving force as likely experienced at the C/10 charging rate, and subsequently propagates spatially at the growth stage. Non-uniform electronic contact can enhance the non-uniformity of reaction but is not a necessary condition. The "surface wetting" terminology could be misleading as wetting is conventionally associated with the surface or interface energy properties of phases. If this is not what the authors intend to relate their observations to, I'll suggest them to avoid using this phrase.

5. A few possible typos: Judging from the context, it seems the word "miscibility" on line 83 and 115 should be replaced by "solubility". Line 125, "(supporting information)" after "Fig. 3a" should be removed.

Three dimensional localization of battery reactions using soft X-ray tomography

Authors' Response

We thank the editor and the three reviewers for the careful consideration about the manuscript. As the reviewer's recommendations, we made significant improvements to the manuscript and present new results to strengthen the technical conclusions and avoid any misunderstandings. In this document, the reviewer comments are *italics*. The changes and referring of the manuscript are in **bolded**.

Reviewer #2

Comment #1:

"The authors have convincingly revised the manuscript following the suggestions of all reviewers. Therefore I recommend publication without further revisions."

Reply #1:

We thank the reviewer for concluding that we have addressed all of the reviewers' recommendations

Reviewer #3

Comment #1:

“This manuscript describes the development of soft X-ray ptychographic tomography and its application to the characterization of phase distribution in partially delithiated LiFePO₄ nanoparticles. The combination of 3D chemical information, a higher spatial resolution than hard x-ray tomography and a larger field of view than electron microscopy makes it a unique and powerful technique, which can potentially provide valuable insights complementary to other existing techniques especially in revealing the physics of battery materials. This work represents a notable progress in method development that in my opinion is potentially worthy of publication in Nature Communications.”

Reply #1:

We thank the reviewer for emphasizing that we have developed X-ray ptychotomography, which is unique and can bring the new insight relevant for battery technology. We believe this work will lead the field to observe the 3-dimensional chemical information that can overcome the limitations of conventional techniques.

Comment #2:

“My concern about the manuscript mainly lies in the interpretation and discussion of the observations made with this technique. The authors present an intriguing finding, namely a trend of increasing active particle population with (010) particle surface area (Fig. 4c) and particle volume (Fig. 4b). They attribute this phenomenon to non-uniform electrical contact between particles and the conductives. It’s stated that “the correlation between large ac facets and increased level of delithiation ... would be a trivial result of the larger probability

of creating multiple points of electrical contact as facet area increases.” I doubt the results presented are sufficient to support this conclusion. The analysis and discussion should be revised and strengthened by taking other factors into consideration. Below are several comments and questions I’d like to ask the authors to address.”

“I’m not fully convinced that the geometric argument made by the authors is the only explanation for the observed size-dependent degree of delithiation. As a counter argument, I want to bring to the authors' attention another possible explanation based on the competition between coherent vs incoherent nucleation. Several recent studies on the hydrogenation of Pd nanoparticles show that coherency stress is responsible for a similar size-dependent phase transformation behavior upon H (de)insertion (see Ulvestad et al. Nature Mater 2017, 16, 565 and references therein). Below a critical size, Pd particles can sustain insertion-induced stress without nucleating defects. The presence of stress increases the nucleation energy barrier and requires a larger driving force (H₂ gas pressure for Pd-H system) to initiate phase transformation. In contrast, particles above the critical size are able to form defects to relax stress, and stress reduction allows phase transformation to proceed at a lower driving force. Can a similar mechanism operate in LiFePO₄, so that a low driving force (overpotential) at the slow C/10 rate can trigger transformation of larger, defect-forming particles but is not sufficient for initiating nucleation in smaller coherent particles? This indeed seems to be consistent with the authors’ earlier work in Ref. 12 and 11 that shows the presence/absence of cracks in large/small LiFePO₄ particles, although the cracked microcrystals studied there are much bigger than the nanoplatelets in this work. It is desirable that the authors can present additional results to prove the correlation or non-correlation between the degrees of delithiation and coherency stress, but at least please comment on this possibility.”

Reply #2:

The hypothesis the reviewer presents is interesting. We start by noting that we observed no clear correlation with particle size, as described by particle volume. Ulvestad *et al.* [Ulvestad, A. *et al.*, *Nature Mater.* **16**, 565-571 (2017)] use this latter magnitude to establish correlations. Our correlations were observed when facet surface area was considered. Furthermore, observations with LiFePO₄ electrodes in the literature would not lend support to the reviewer's hypothesis. Generally speaking, it is very well established that LiFePO₄ nanoparticles (100 nm or less) lead to electrodes showing better capabilities for fast rate at the same overpotential than particles an order of magnitude larger (*i.e.*, submicron to micron). In other words, larger particles require higher overpotentials to drive the reaction than smaller ones, and, in some cases, do not even proceed. The large particles in our refs [Shapiro, D. A. *et al.*, *Nature Photon.* **8**, 765-769 (2014); Yu, Y.-S. *et al.*, *Nano Lett.* **15**, 4282-4288 (2015)] are not functional in the same electrochemical conditions that we used for the nanoplates in this study. The better electrochemical response of nanoparticles is driven by the activation of a metastable solid solution pathway. In this pathway, nucleation and growth is bypassed to avoid surface energy penalties, but diffusion induces a gradient in compositions that rapidly triggers spinodal decomposition and leads to mixtures of end members in the timescale of ms [Abdellahi, A. *et al.*, *J. Mater. Chem. A*, **4**, 5436 (2016)]. This mechanism is consistent with the *ex situ* observations made here. As far as we understand from the explanation above, this heterogeneous pathway is related to the mechanism that the reviewer proposes for the incorporation of H in Pd. However, formation of these solid solution intermediates is associated with low energy penalties, just around 15 meV/formula unit, as described in the

literature [Malik, R. *et al.*, *Nature Mater.* **10**, 587-590 (2011)]. Therefore, they are not large enough to preclude the reaction in the way that the reviewer proposes.

Comment #3:

“I find the large active particle population seen here compared to other systems (e.g. Chueh’s work) is consistent with the presence of large stress in the small nanoplates as coherency stress can cause the energy barrier for phase growth to increase with SOC and thus prevent the particle from complete transformation. Therefore a discussion on the potential effect of stress appears to be necessary.”

Reply #3:

We wish to clarify that these particles are capable of completely transforming from LiFePO_4 to FePO_4 . Discussions of the effect of stress in this system are abundant in the literature [Malik, R. *et al.*, *Nature Mater.* **10**, 587-590 (2011)]. We have summarized the salient points above, and re-emphasize that small particle dimensions activate a solid solution pathway that minimizes strain. For reasons of conciseness, we prefer to avoid restating facts in the paper that are already extensively discussed by others. Nonetheless, we recognize that this driving force is not specifically called out in our discussion. As a result, we added the following statement in the introduction (see highlighted text in page 4) which maps out the role of strain/stress guiding the mechanism of phase transformation when particle size is reduced.

These metastable pathways are triggered by the fact that nucleation of a second phase becomes unfavorable below a certain particle size because of the scaling relationship of the critical nucleation barrier with the increasing surface energy. Mixing energies associated with these solutions are calculated to be low²¹, indicating that low overpotentials are enough to

enable fast charging in a regime that is only controlled by carrier diffusion, as opposed to movement of domain walls. However, carrier diffusion still leads to compositional gradients within an individual particle during charging, which would, as a result, rapidly undergo spinodal decomposition to mixtures of the end members even as the reaction is proceeding²⁵.

Comment #4:

“Despite the statement “The apparent discrepancy between this value and the 50% SOC of the electrochemical cell is addressed below” on page 5, the authors do not seem to address this point head-on in the manuscript. A more complete discussion is needed. Reading between the lines, I think the authors suggest the reason for the discrepancy also lies in the dependence of electrical contact area on particle size. However, to support this claim, a plot of particle SOC vs the (010) facet area will be useful to demonstrate the trend instead of the distribution of active particle fraction shown in Fig. 3. In addition, can the authors exclude the possibility that the local electrode region where the cluster of nanoplate particles were imaged happened to have low carbon content and be poorly electronically wired?”

Reply #4:

We did indeed discuss the discrepancy a few paragraphs below the statement (page 7): “The fact that particles with small facets were both systematically found to be closer to the reduced state (Fig. 3c) and oversampled in the tomogram compared to the overall population of the electrode (Fig. S13) helps explain the discrepancy between the overall SOC in the 3D image and the electrochemical cell.” That said, it is perfectly possible that, as the reviewer states, the region from where particles were harvested happened to have lower carbon contents than the rest of the electrode. To fulfill reviewer’s concern, we added the following statements (highlighted sentences in page 7).

It is also possible that the population of particles harvested for the tomogram were located in a portion of the electrode with a deficiency in carbon content and/or electrolyte wetting, introducing transport deficiencies that delayed their reaction.

Comment #5:

“On page 9, the authors state that “The existence of multiple points of reaction initiation within a particle can only be explained by a recently proposed model where surface wetting was considered in addition to coherency strain and elastic moduli within nanoparticles of similar dimensions.” I question the need and value of invoking the “surface wetting” concept here. The local initiation of reaction on surface could simply be understood as heterogeneous nucleation events, which is the first step in the nucleation and growth process. Even if the particle surface has a uniform electronic contact with carbon and electrolyte network, reaction will still happen locally on surface (i.e. nucleation) at a low driving force as likely experienced at the C/10 charging rate, and subsequently propagates spatially at the growth stage. Non-uniform electronic contact can enhance the non-uniformity of reaction but is not a necessary condition. The “surface wetting” terminology could be misleading as wetting is conventionally associated with the surface or interface energy properties of phases. If this is not what the authors intend to relate their observations to, I’ll suggest them to avoid using this phrase.”

Reply #5:

Our purpose in the section that the reviewer cited is comparison of our observations with all the possible mesoscale models of lithiation of LiFePO_4 available in the literature. We discuss their relationship with our observations extensively and find that the model by Welland *et al.* [Welland, M. J. *et al.*, *ACS Nano* **9**, 9757-9771 (2015)] approaches them the

most. The main difference we noticed with other models is the inclusion of the concept of surface wetting, which we interpret, after private discussions with Welland *et al.*, as consistent with heterogeneity in electrical contacts. The reviewer states that “*reaction will still happen locally on surface (i.e. nucleation) at a low driving force as likely experienced at the C/10 charging rate, and subsequently propagates spatially at the growth stage*”. However, as described, this would lead to a core-shell model, which is inconsistent with our observations. Other models based on the anisotropy in ion diffusion and lattice misfit [Malik, R. *et al.*, *J. Electrochem. Soc.* **160**, A3179 (2013)] also produce pictures that fail to match our observations. We are not aware of other models based on nucleation and growth that would reproduce the complex patterns of delithiation in our measurements absent heterogeneity introduced by gradients in the supply of electrical contacts.

Comment #6:

“A few possible typos: Judging from the context, it seems the word “miscibility” on line 83 and 115 should be replaced by “solubility”. Line 125, “(supporting information)” after “Fig. 3a” should be removed.”

Reply #6:

Following the reviewer’s comment, “miscibility” was changed to “solubility”. In addition, typos were corrected throughout the manuscript.

Reviewers' comments:

Reviewer #3 (Remarks to the Author):

The authors' responses and the revised manuscript have satisfactorily addressed some of this reviewer's comments. But I have to admit I still have reservation about their responses to comment #2 and #5, and so would like to make the same suggestions again.

On reply #2:

* I want to make it clear that I agree with the authors that heterogeneous electrical contact provides a plausible explanation to their observation, but also believe that the effect of coherent stress offers an alternative interpretation, which cannot be ruled out by the results presented in the manuscript. A more balanced discussion is needed.

* The authors' viewpoint is that because nanosized LiFePO₄ delithiates through the metastable solid solution that minimizes misfit strain, coherent stress should not play a significant role in determining the phase transformation pathway. I want to point out that the reason that the equilibrium first-order phase transition is bypassed in nanoscale LiFePO₄ is exactly because coherent stress suppresses phase separation (e.g. see Cogswell and Bazant ACS Nano 2012, 6, pp. 2215). The reason that metastable solid solution doesn't form in larger LiFePO₄ particles is that dislocations and cracks can form above a critical particle size to relax misfit stress, which is evidenced by many experiments including those by the authors. Coherent stress thus does play an important role.

* Even when delithiation initially occurs through metastable solid solution, after charging is terminated, partially delithiated small particles could re-lithiate to reduce the coherent strain energy stored inside the particles. This is a phenomenon that has been frequently suggested in literature and may contribute to the observation reported here.

* The response states that "nucleation and growth is bypassed to avoid surface energy penalties". This statement is not correct. Nucleation is bypassed because of the presence of misfit strain for LiFePO₄ as shown by the work of Cogswell cited above.

* The authors suggest that the fact that "larger particles require higher overpotentials to drive the reaction than smaller ones" points to that nucleation should not be more difficult in small particles. It needs to be distinguished here that the overpotential for phase transition consists of two contributions, i.e. those for overcoming nucleation and growth, respectively. Large particles have a small overpotential for nucleation, but requires a larger overpotential for the growth of the new phase because the interfacial defects make phase boundary movement more difficult. On the other hand, small particles requires a larger overpotential for nucleation due to coherent stress, but once the new phase is nucleated it requires very small overpotential to move the coherent interface.

* The response also mentions that the calculated energy barrier for metastable solid solution formation at ~15 mV is very small. However, this calculation is done assuming uniform concentration in the lattice. When a Li concentration gradient exists in the particle due to kinetic limitation under realistic charging conditions, the misfit strain arising from composition inhomogeneity will cause a higher energy barrier.

Regarding Reply #5, I want to re-iterate my concern is that the use of terminology "surface wetting" is misleading. Wetting specifically refers to the phenomenon of surface energy favoring one phase over the other in the system. It facilitates phase transformation by providing additional

thermodynamic driving force. On the other hand, the effect of non-uniform electrical contact is a kinetic one and not related to surface energy as considered in the work of Welland et al. (ACS Nano 2015). The term "wetting" is used in a very loose sense here and may cause confusion. Why not just say "heterogeneous electrical contact results in local nucleation hotspots"?

On Reply #3, the revised text states "...nucleation of a second phase becomes unfavorable below a certain particle size because of the scaling relationship of the critical nucleation barrier with the increasing surface energy". As mentioned above, current understanding of LiFePO₄ attributes the suppression of the nucleation of the second phase to coherent strain energy, and this sentence needs to be modified.

Three dimensional localization of battery reactions using soft X-ray tomography

Authors' Response

The reviewer comments are in *italics*. The changes and referring of the manuscript are in **bold**.

Reviewer #3

Comment #1:

“The authors’ responses and the revised manuscript have satisfactorily addressed some of this reviewer’s comments. But I have to admit I still have reservation about their responses to comment #2 and #5, and so would like to make the same suggestions again. “

“I want to make it clear that I agree with the authors that heterogeneous electrical contact provides a plausible explanation to their observation, but also believe that the effect of coherent stress offers an alternative interpretation, which cannot be ruled out by the results presented in the manuscript. A more balanced discussion is needed.”

Reply #1:

We thank the reviewer for concluding that we have addressed the previous comments by the reviewers and for agreeing with our proposal of heterogeneous electrical contacts. We wish to note that there do not appear to be any reservations regarding the technical quality and value of the developments in this work, and, rather, disagreements are centered on the issue of discussion of the observations in light of existing models of phase transformation. We have

further restructured the manuscript with the goal of addressing his/her comments regarding the “balance” of this discussion of results. For the specific case of the role of strain energetics on the correlation between the size of the *ac* facet and the extent of delithiation, we wish to highlight a new discussion of the existing models that calculate barriers for phase coexistence in LiFePO₄ nanoparticles (page 9), describing the conflicting results in the literature:

In a scenario where reaction progression is conditioned by electrical contacts, the correlation between large *ac* facets and increased level of delithiation found over many particles, also suggested by others⁴⁶, could result from the larger probability of creating multiple points of electrical contact as facet area increases. Thermodynamic origins, imposed by coherency strain along the *ac* facet, are also possible. Calculations of energy barriers to phase co-existence in a Li_xFePO₄ nanoparticle available in the literature are conflicting. The 3D model by Welland *et al.* led to an inverse relationship with size⁴⁶, which would predict larger nanoparticles reacting first, as observed here. In contrast, Cogswell *et al.* found lower or comparable barriers with decreasing size using a 2D model of the *ac* facet with depth averaging⁴⁹. The discrepancy could, at least in part, be due to the different interfacial orientations in the two models, suggesting that further computational work would be helpful to establish this question.

Comment #2:

“The authors’ viewpoint is that because nanosized LiFePO₄ delithiates through the metastable solid solution that minimizes misfit strain, coherent stress should not play a

significant role in determining the phase transformation pathway. I want to point out that the reason that the equilibrium first-order phase transition is bypassed in nanoscale LiFePO₄ is exactly because coherent stress suppresses phase separation (e.g. see Cogswell and Bazant ACS Nano 2012, 6, pp. 2215). The reason that metastable solid solution doesn't form in larger LiFePO₄ particles is that dislocations and cracks can form above a critical particle size to relax misfit stress, which is evidenced by many experiments including those by the authors. Coherent stress thus does play an important role.”

“On Reply #3, the revised text states “...nucleation of a second phase becomes unfavorable below a certain particle size because of the scaling relationship of the critical nucleation barrier with the increasing surface energy”. As mentioned above, current understanding of LiFePO₄ attributes the suppression of the nucleation of the second phase to coherent strain energy, and this sentence needs to be modified.”

“The response states that “nucleation and growth is bypassed to avoid surface energy penalties”. This statement is not correct. Nucleation is bypassed because of the presence of misfit strain for LiFePO₄ as shown by the work of Cogswell cited above.”

Reply #2:

Contrary to what the reviewer states, the fact that coherency strain is a determining factor on the opening of a solid solution pathway for delithiation, or that it dominates the transformation of Li_xFePO₄ more broadly, is not disputed anywhere in the article, including the quoted statement. It is indeed extensively reported in the literature, to which we refer the reader for conciseness. The statements in our previous reply indicated that, once a solid solution is formed, issues of coherency strain become much less prominent than during a classical nucleation and growth mechanism, not that they do not determine to the specific mechanism.

We invite the reviewer to revisit reference 21, which we used as basis for the text the reviewer quotes, for a more extensive discussion of the interplay between coherency strain and surface energy to unlock the solid solution pathway. Recognizing that the choice of words may have been incomplete, and to avoid any ambiguity on the prominent role of coherency strain in the transformation of Li_xFePO_4 , we have modified statements in the introduction (page 4), added the statements quoted in Reply #1 and below (page 9), and added a new reference (current ref. 47; Cogswell and Bazant, *ACS Nano* **2012**, 6, 2215). The newly added and modified statements clearly map out the role of coherency strain/stress dominating the phase transformation, and eliminate a reference to surface energy effects, which require nuance that is beyond the scope of the manuscript. We hope the reviewer will be satisfied with these modifications, which we believe fully address his/her concerns.

These pathways bypass penalties in coherency strain due to the co-existence of phases in one particle, both enabling completion of the reaction and faster kinetics.

If transient Li_xFePO_4 solid solutions were present under working conditions because phase separation is suppressed by coherency strain in these small particles^{21,47}, a heterogeneous distribution of electrical contacts, and, thus, carrier diffusion, could impose gradients in x within a particle. Such gradients were recently predicted to trigger immediate spinodal decomposition into two phases even under applied current⁴⁸, which would be consistent with our *ex-situ* observations.

Comment #3:

“Even when delithiation initially occurs through metastable solid solution, after charging is terminated, partially delithiated small particles could re-lithiate to reduce the coherent strain energy stored inside the particles. This is a phenomenon that has been frequently suggested in literature and may contribute to the observation reported here.”

Reply #3:

We start by noting that there are no statements in our manuscript claiming information on dynamics. The issue of relaxation from a kinetic solid solution is addressed in the article, by comparison to the models of relaxation that have been reported in the literature (R. Malik, F. Zhou and G. Ceder, *Nature Mater.* **2011**, 10, 587; M. Wagemaker, W. J. H. Borghols and F. M. Mulder, *J. Am. Chem. Soc.* **2007**, 129, 4323; W. Dreyer *et al.*, *Nature Mater.* **2010**, 9, 448). When the electrochemical stimulus is removed, metastable solid solutions equilibrate to the thermodynamically favored two-phase separation. However, precisely due to the unfavorable formation of boundaries within a particle when it is very small (as proposed, among others, in ref. 21 and by Wagemaker, *et al. J. Am. Chem. Soc.* **2007**, 129, 4323), such separation is predicted to involve interparticle exchange of Li, so that the end result is a mixture of particles that are composed of either pure LiFePO₄ or pure FePO₄. Experimental observations exist of such distribution (first, in Delmas *et al. Nat. Mater.* **2008**, 7, 665 and, later, in Chueh *et al. Nano Lett.* **2013**, 13, 866), which Malik *et al.* (ref. 21) argued are consistent with their models. It is not the case in our observations, where two-phases co-existed within a particle, so such relaxation mechanisms are unlikely to have occurred as proposed. In contrast, recently, Abdellahi *et al.* proposed that gradients between the surface and the interior of a particle in a transient Li_xFePO₄ solid solution could create a driving force for immediate spinodal decomposition even under applied current, not requiring the reaction to be stopped (Abdellahi *et al. J. Mater. Chem. A* **2016**, 4, 5436). This model could explain our observations, and we have stated as much in the manuscript. We have extensively

reorganized the discussion in this revised version in pages 8 and 9 to further emphasize these points.

The 3D snapshot of partly reacted states revealed the existence of sharp inhomogeneities within a particle, ascribed to co-existence of $\text{Li}_\alpha\text{FePO}_4$ or $\text{Li}_\beta\text{FePO}_4$, where $\alpha \gg \beta$ and $(\alpha - \beta)$ defines the miscibility gap. The exact values of α and β could not be extracted at the chemical resolution of our measurement. Phase co-existence has been observed in nanoparticles by others^{12,18,37,38}. It has been shown to prevail during slow charging in recent *operando* X-ray studies^{36,39,40}. This intra-particle heterogeneity, in *ex-situ* conditions, is inconsistent with a models of domino-cascade delithiation¹⁹ or relaxation from a kinetic solid solution state through interparticle exchange of Li ^{21,41,42}, where populations of particles containing only either $\text{Li}_\alpha\text{FePO}_4$ or $\text{Li}_\beta\text{FePO}_4$ would be expected.

The distribution of delithiated domains in individual particles was complex and irregular irrespective of their individual Li content. Complex patterns of delithiation in LiFePO_4 nanoparticles have been recently predicted by Welland *et al.* using a thermodynamic 3D model where the effect of coherency strain, elastic moduli and surface wetting was considered⁴⁶. Kinetically, such patterns could be aggravated by the existence of uneven electrical contacts within individual nanoparticles in a composite porous electrode, driven by local heterogeneities in

component distribution. These contacts act as sinks of ions and electrons, and, thus, points of reaction initiation.

Comment #4:

“The authors suggest that the fact that “larger particles require higher overpotentials to drive the reaction than smaller ones” points to that nucleation should not be more difficult in small particles. It needs to be distinguished here that the overpotential for phase transition consists of two contributions, i.e. those for overcoming nucleation and growth, respectively. Large particles have a small overpotential for nucleation, but requires a larger overpotential for the growth of the new phase because the interfacial defects make phase boundary movement more difficult. On the other hand, small particles requires a larger overpotential for nucleation due to coherent stress, but once the new phase is nucleated it requires very small overpotential to move the coherent interface.”

Reply #4:

We take the point made by the reviewer. There are no statements making this point in the manuscript.

Comment #5:

“The response also mentions that the calculated energy barrier for metastable solid solution formation at ~15 mV is very small. However, this calculation is done assuming uniform concentration in the lattice. When a Li concentration gradient exists in the particle due to kinetic limitation under realistic charging conditions, the misfit strain arising from composition inhomogeneity will cause a higher energy barrier.”

Reply #5:

We invite the reviewer to evaluate the work by Abdellahi *et al.* (*J. Mater. Chem. A* **2016**, 4, 5436). As far as we can tell, they calculated the barriers in instances like the one the reviewer brings up. The barriers were still found to be small, which explains why the solid solution pathway can still prevail in the presence of such intraparticle gradients imposed by diffusion.

Comment #6:

*“Regarding Reply #5, I want to re-iterate my concern is that the use of terminology “surface wetting” is misleading. Wetting specifically refers to the phenomenon of surface energy favoring one phase over the other in the system. It facilitates phase transformation by providing additional thermodynamic driving force. On the other hand, the effect of non-uniform electrical contact is a kinetic one and not related to surface energy as considered in the work of Welland *et al.* (ACS Nano 2015). The term “wetting” is used in a very loose sense here and may cause confusion. Why not just say “heterogeneous electrical contact results in local nucleation hotspot”?”*

Reply #6:

We agree with the reviewer that the term can lead to confusion, as it has a thermodynamic origin. We have modified the subsequent statements in the article to focus on the fact that the domain distributions proposed by Welland *et al.* are those that most closely resemble our observations by spectrotomography.

REVIEWERS' COMMENTS:

Reviewer #3 (Remarks to the Author):

The revised manuscript addresses my comments. I recommend its acceptance.

Three dimensional localization of battery reactions using soft X-ray tomography

Authors' Response

The reviewer comments are in *italics*.

Reviewer #3

Comment #1:

"The revised manuscript addresses my comments. I recommend its acceptance.."

Reply #1:

We thank the reviewer for concluding that we have addressed all of the reviewers' recommendations